# Bridging Bots: from Perception to Action via Multimodal-LMs and Knowledge Graphs

**Margherita Martorana**  M.MARTORANA@VU.NL

**Francesca Urgese**  F.URGESE@STUDENT.VU.NL

**Mark Adamik**  M.ADAMIK@VU.NL

**Ilaria Tiddi**  I.TIDDI@VU.NL

*Vrije Universiteit Amsterdam, The Netherlands*

**Editors:** Leilani H. Gilpin, Eleonora Giunchiglia, Pascal Hitzler, and Emile van Krieken

## Abstract

Personal service robots are increasingly deployed to support daily living in domestic environments, particularly for elderly and individuals requiring assistance. These robots must perceive complex and dynamic surroundings, understand tasks, and execute context-appropriate actions. However, current systems typically rely on proprietary, hard-coded solutions tied to specific hardware and software, resulting in siloed implementations that are difficult to adapt and scale across platforms. Ontologies and Knowledge Graphs (KGs) offer a solution to enable interoperability across systems, through structured and standardized representations of knowledge and reasoning. However, symbolic systems such as KGs and ontologies struggle with raw and noisy sensory input. In contrast, multimodal language models are well suited for interpreting input such as images and natural language, but often lack transparency, consistency, and knowledge grounding. In this work, we propose a neurosymbolic framework that combines the perceptual strengths of multimodal language models with the structured representations provided by KGs and ontologies, with the aim of supporting interoperability in robotic applications. Our approach generates ontology-compliant KGs that can inform robot behavior in a platform-independent manner. We evaluated this framework by integrating robot perception data, ontologies, and five multimodal models (three LLaMA-based and two GPT-based models), each using different modes of neural-symbolic interaction. We assess the consistency and effectiveness of the generated KGs across multiple runs and configurations, and perform statistical analyzes to evaluate performance. Results show that GPT-o1 and LLaMA 4 Maverick consistently outperform other models. However, our findings also indicate that newer models do not guarantee better results, highlighting the critical role of the integration strategy in generating ontology-compliant KGs.

## 1. Introduction

Service robots are increasingly deployed in real-world environments to assist with everyday tasks and provide support to older people and individuals with disabilities Macis et al. (2023); Nanavati et al. (2024); Holland et al. (2021); Roy et al. (2000); Sørensen et al. (2024). Despite significant progress, most systems remain limited to narrowly defined scenarios: preprogrammed combinations of tasks, environments, and interactions tailored to specific hardware and software. This rigidity stems from siloed, hard-coded architecture unique to individual robot models or platforms, which lack generalizability and make it difficult

to adapt, reuse, or transfer capabilities across different systems and application domains García et al. (2023); Wang et al. (2024).

To support interoperability in robotic applications, a possible solution is the use of ontologies and Knowledge Graphs (KGs), which provide structured and standardized representations of knowledge that enable reasoning, integration and knowledge sharing. Symbolic approaches such as these are well suited for platform-independent applications and the reuse of information across systems fPaulius and Sun (2019). However, ontologies and KGs often struggle with the unstructured and noisy nature of real-world sensory input. Robots operating in everyday environments must often interpret raw sensory data such as images and speech - inputs that symbolic systems alone struggle to process effectively Langley et al. (2009). In contrast, multimodal language models are effecting at processing such data, but often lack transparency, consistency and knowledge grounding, limiting their suitability for structured and reusable reasoning Zhang et al. (2021). To address these complementary strengths and limitations, this paper explores a neurosymbolic integration framework that combines the perceptual capabilities of multimodal models with the structured, interoperable representations provided by ontologies and KGs. We investigate how these components can work together to generate ontology-compliant KGs from sensory input, to ultimately support robotic reasoning and behavior in a platform-independent and reusable manner.

Let us consider the following scenario in which a service robot might operate:

> A robot operating in a household kitchen is tasked with organizing, tidying, and cleaning. To perform these tasks effectively, it must perceive and interpret its surroundings, identify relevant objects, and plan an appropriate sequence of actions. For example, it may: 1) recognize and differentiate objects such as plates, utensils, and appliances; 2) decide on the appropriate sequence of actions needed to complete the task; 3) interact with objects and appliances; 4) restore order in the environment.

Here, the robot must convert raw sensory, such as images, input into meaningful, structured knowledge that supports action planning and reasoning: first, picking up a plate, then opening the dishwasher, then selecting the appropriate wash cycle. This process must be adaptive to the tasks the robot has to perform within the environment, and cannot rely on rigid, platform-specific logic Paulius and Sun (2019). Instead, it requires a generalizable and adaptable approach that can function across different robotic architectures and handle dynamic real-world environments.

To guide our investigation in how neurosymbolic methods can be used to generate interoperable representations of environments and actions, we formulate the following question:

### How can neural models and symbolic representations be effectively combined to generate structured, context-aware representations of environments and action sequences for service robots?

We propose and evaluate several neurosymbolic approaches that integrate the perceptual strengths of multimodal language models - which are well suited for interpreting raw sensory input such as images and natural language - with the structured reasoning capabilities of ontologies and KGs. By combining these paradigms, we explore how robots can construct robust, context-aware knowledge of their environment and generate action sequences that are interoperable and transferable across platforms.

## 2. Background

Robots operating in real-world environments require flexible, context-aware knowledge representations Paulius and Sun (2019). However, effectively integrating symbolic and neural reasoning remains a challenge, especially in environments characterized by dynamic and layered tasks Langley et al. (2009). Effective deployment of assistive service robots requires internal knowledge representations that are not only rich but also flexible enough to generalize across varying use contexts Roy et al. (2000). The need for such flexible and interpretable frameworks is particularly evident in human-centered domains like healthcare, where service robots must communicate and act transparently to gain user trust Holland et al. (2021). Expectations from end-users further underscore this need: care-receivers value humanoid robots that are predictable, responsive, and understandable in their behavior Sørensen et al. (2024), and studies on smart home robotics reveal that adaptability and transparency are key to user satisfaction Wang et al. (2024).

### 2.1. From Symbolic Representations to Action Generation

Ontologies have become essential for structuring robotic knowledge, allowing systems to represent objects, actions, and environments in a formal and interpretable way. Standards like IEEE Robotics and Automation Society (2015, 2021) support semantic interoperability across platforms, though they often lack task-level detail and are not integrated with perception. Donadello and Serafini (2016) pioneered semantic image interpretation by aligning segmented images with ontological models using a loss function over spatial and symbolic structure, grounding perception into logic. Several frameworks have since attempted to bridge perception and symbolic knowledge. Olivares-Alarcos et al. (2022) models human-robot collaboration and adaptive plans, but with a focus on abstract schemas, without supporting automated population from sensory input. Ge et al. (2024) combines ontologies with object detection, yet remains constrained to fixed domestic tasks. In manufacturing, symbolic systems have been used for action planning and verification Balakirsky and Kootbally (2015), but usually assume static environments. Earlier work by Chandrasekaran et al. (1998) proposed formal task decomposition models, though without perceptual grounding. More recent proposals Schlenoff et al. (2017) highlight the need for executable task representations, but integration with learning remains limited.

Symbolic planning frameworks such as PDDL Aeronautiques et al. (1998) have long supported task execution by specifying preconditions, effects, and goals. Recent work has aimed to improve their flexibility by connecting them to ontologies. For instance, ORKA Adamik et al. (2024) provides a reusable ontology to support knowledge acquisition pipelines, emphasizing modularity and generalization across settings. While planning remains an important goal, our focus lies on an earlier step: producing structured, symbolic representations from raw perceptual input. That is, our approach builds on prior efforts by automating ontology population from perception and focusing on reusable, grounded symbolic knowledge. This aligns with the FAIR principles for data management Wilkinson et al. (2016), with the aim of producing interoperable and reusable outputs that support integration across systems.

## 2.2. Visual Language Models in Robotics

Visual Language Models (VLMs) are being adopted to help robots interpret scenes and instructions with both vision and language. Models like Driess et al. (2023) and Brohan et al. (2023) can combine perception with high-level reasoning, but their outputs can be hard to interpret and difficult to reuse. Recent work has applied VLMs to generate semantic task graphs Zhao et al. (2024) or interpret multimodal commands Wu et al. (2024), yet these systems still struggle with structured reasoning and symbolic consistency. LLMs have also been used to populate ontologies from natural language Li et al. (2024), offering a path toward symbolic grounding. In practice, however, most of these models are difficult to access or deploy on physical robots. Their size makes local use infeasible, while remote APIs often come with usage limits, compatibility issues, or reliability concerns. These constraints have a significant impact on model selection and system design in real-world settings. Our approach addresses these challenges by transforming neural outputs into symbolic representations, i.e. KGs. We experiment with different configurations, focusing on how well robots can extract relevant elements from a scene and structure, and use them to generate valid, context-aware actions.

## 3. Proposed Approach

We propose a neurosymbolic framework that integrates raw sensory input — multi-view RGB images extracted from the Webots[1] simulation platform, using a 3D kitchen environment — task instructions, and ontological knowledge to generate structured, interpretable representations of the robot's environment and actions. These representations take the form of two KGs: an observation graph Adamik et al. (2024), which captures the current state of the environment, and an action graph, which encodes the sequence of actions needed to complete a given task. This section provides details on our framework, the models and methods used to generate the KGs, and the design of our experimental evaluation.

## 3.1. Neurosymbolic Setup

Our pipeline (see Figure 1) integrates symbolic and neural components to produce ontology-compliant KGs from multimodal inputs. Specifically, it combines three inputs:

1. **Raw sensory data:** a set of images captured from the Webots simulation platform, showing a domestic kitchen from multiple angles. These images provide a comprehensive visual overview of the scene and are used to inform the robot's perception of the layout and objects in the environment.

2. **Task description:** a natural language instruction specifying the robot's objective. In our case: *"Restore the kitchen to an organized state by identifying all misplaced items and returning them to their standard storage locations based on their type and function. Prioritize actions according to logical task order, and perform each step atomically."*

---

1. https://cyberbotics.com

3. **Ontology:** the *Ontology for roBOts and acTions* (OntoBOT)[2] provides a formalized and reusable structure for representing objects, properties, spatial relationships, and actions. It serves as a grounding structure to standardize the perception of both the environment and actions. The ontology is provided in the form of RDF triples, serialized in Turtle format.

**Multimodal Language Models** We experiment with a range of multimodal language models capable of interpreting these inputs: *LLaVA + LLaMA 3*, a modular combination we constructed for experimental purposes, where visual understanding (via LLaVA) and language processing (via LLaMA 3) are decoupled; *LLaMA 4 Scout and Maverick*, newer LLaMA-based models with native multimodal capabilities that jointly process images and text in an end-to-end fashion - Scout is optimized for long-context tasks, while Maverick uses more parameters and is designed for strong general-purpose performance; *GPT 4.1-nano and o1*: OpenAI's end-to-end multimodal models, capable of integrating visual and textual reasoning - o1 is larger but slow and computationally intensive, while 4.1-nano offers faster performance with a smaller model size. This selection allows us to compare modular vs. unified architectures and assesses how design differences influence the quality of symbolic outputs. As discussed in Section 2.2, models are accessed via remote APIs, to reflect realistic deployment constraints on robotic systems. Specifically, we accessed LLaVA through Nebula[3], LLaMA variants via GroqCloud[4] and LLaMA-index[5], and GPT models through OpenAI's API[6].

## 3.2. KG Generation

As mentioned earlier, our framework generates two distinct KGs from the input data: an observation graph and an action graph (see Figure 1). Each graph is created using one of four neurosymbolic methods, which differ in how they integrate perceptual input and ontological structure. To generate the observation graph, we evaluate two strategies:

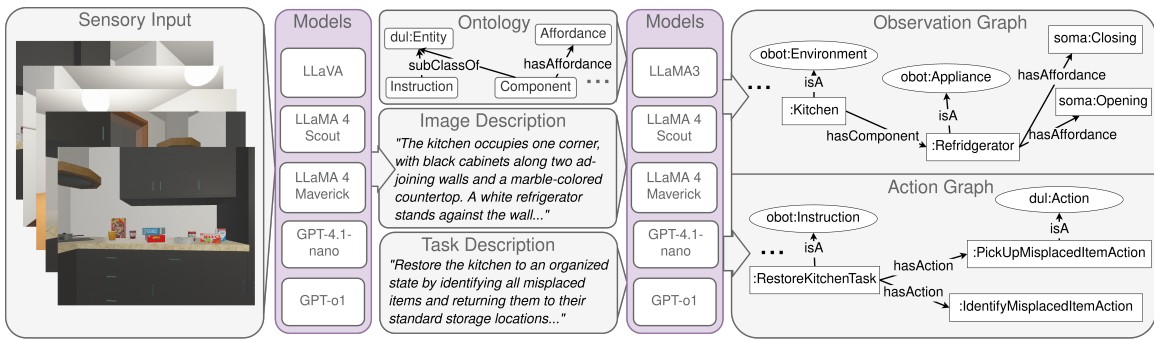

Figure 1: Overview of the neurosymbolic pipeline.

2. https://w3id.org/onto-bot
3. https://networkinstitute.org/nebula/
4. https://console.groq.com/keys
5. https://www.llamaindex.ai
6. https://openai.com/api/

**1. Narrative-based generation:** The model first generates a natural language description of the environment based on the input image. This narrative is then used, together with the ontology, to construct the KG. This strategy is used in:

- **Dynamic Path Extractor (DPE)**: extracts structured paths aligned with the ontology from the generated description. DPE is only available for LLaMa models.

- **Description to Knowledge Graph (D2KG)**: directly generates a symbolic graph from the textual description and ontology provided in the prompt.

- **D2KG with Retrieval-Augmented Generation (D2KG-RAG)**: similar to D2KG, but the ontology is retrieved at inference time from a vector database.

**2. Vision-based generation:** In this approach, the model is prompted with the image and ontology together, bypassing the textual description entirely. The observation graph is produced directly via:

- **Image to Knowledge Graph (I2KG)**: maps visual input into symbolic form using ontology guidance.

With the observation graph in place, we generate the action graph using the same four strategies, adapted to also include the task description. Each strategy processes: the scene (described textually or visually, as above); the ontology (embedded in the prompt or accessed via RAG); and a natural language task description. The output is a KG describing the sequence of actions the robot has to perform to complete the task. As before, **DPE**, **D2KG**, and **D2KG-RAG** operate on textual inputs, while **I2KG** maps directly from images and task prompts to structured actions.

### 3.3. Evaluation

To evaluate the framework, we conducted a series of experiments for each model and configuration. For every setup, we generate KGs over 10 independent runs, resetting the session each time to avoid any information carryover. This allowed us to assess both the average performance and consistency of the outputs. To evaluate the quality and structure of the generated KGs, we apply three evaluation strategies aim to capture the syntactic and ontological correctness, and statistical significant across models and methods.

**KG Quality Metrics** This evaluation tracks general statistics and compliance properties of the generated KGs. For each neurosymbolic method, we generate the KGs 10 times and compute the following metrics: *RDF validity*, the proportion of generated KGs that are valid RDF in Turtle (TTL) format; *triple count*, the average number of triples per graph, providing a sense of graph richness and details; *ontology compliance*, the percentage of properties and classes in each KG that are consistent with those defined in OntoBOT; and *ontology coverage*, the proportion of properties and classes from OntoBOT that appear at least once in the KG.

**SHACL-Based Structural Validation**   To further assess structural and ontological correctness, we employ the Shapes Constraint Language (SHACL)[7] for rule-based validation. Each generated graph is tested against a set of SHACL shapes based on the OntoBOT ontology, verifying whether the graph adheres to expected structural patterns and constraints (e.g. required properties, class-property relationships). The SHACL shapes used for validation are provided in Appendix B.

**Statistical Comparison**   To assess differences across models and methods, we conduct statistical significance testing using the Mann-Whitney U test. We apply this test to measure whether distributions of compliance and coverage scores differ significantly across models and neurosymbolic integration methods. These analyses provide insight into whether certain neurosymbolic integration strategies consistently outperform others in terms of ontological fidelity and structural soundness. All code, data, prompts and results produced in our experiments are publicly available on GitHub[8]. Illustrative examples of the prompts used and generated KGs are also provided in Appendix A and C, respectively.

## 4. Results

In this section, we present the results of our evaluation across the different models and integration methods. Each experiment was repeated 10 times to account for variability in model outputs. All generated KGs from the models are available in the project GitHub[9], with two representative examples included in Appendix C. Additionally, all data used for statistical analyses, as well as the analysis scripts themselves, are also online[10].

### 4.1. KG Quality Metrics and Structural Validation

Firstly, we assess the structural validity and syntactic quality of the generated KGs. Figure 2 shows compliance and coverage metrics for both properties and classes, across models and graph types, i.e. observation graphs (solid fill) and action graphs (hatched fill).

LLaMA 4 Maverick achieves the highest compliance and coverage scores across both graph types in most cases, followed closely by GPT-o1. In contrast, GPT-4.1-nano performs significantly worse, with most metrics falling below 30%. Interestingly, in most cases action graphs tend to show slightly higher compliance than observation graphs. This indicates that including the task description in the prompt does not hinder model performance in generating ontology-compliant KGs. Unified multimodal models (e.g. LLaMA 4 Scout and Maverick) generally outperform the modular `LLaVa + LLaMa 3` pipeline. This trend supports the idea that end-to-end vision-language integration improves structured output generation when guided by an ontology schema. High standard deviations - particularly in outputs from GPT-4.1-nano, LLaMa Scout and LLaVa + LLaMa 3 - highlight variability in KG quality across runs. While Maverick and GPT-o1 are more stable, they still show variations, suggesting output consistency remains a challenge even among top performers.

---

7. https://www.w3.org/TR/shacl/

8. https://github.com/kai-vu/bridging-bots

9. https://github.com/kai-vu/bridging-bots

10. https://github.com/kai-vu/bridging-bots

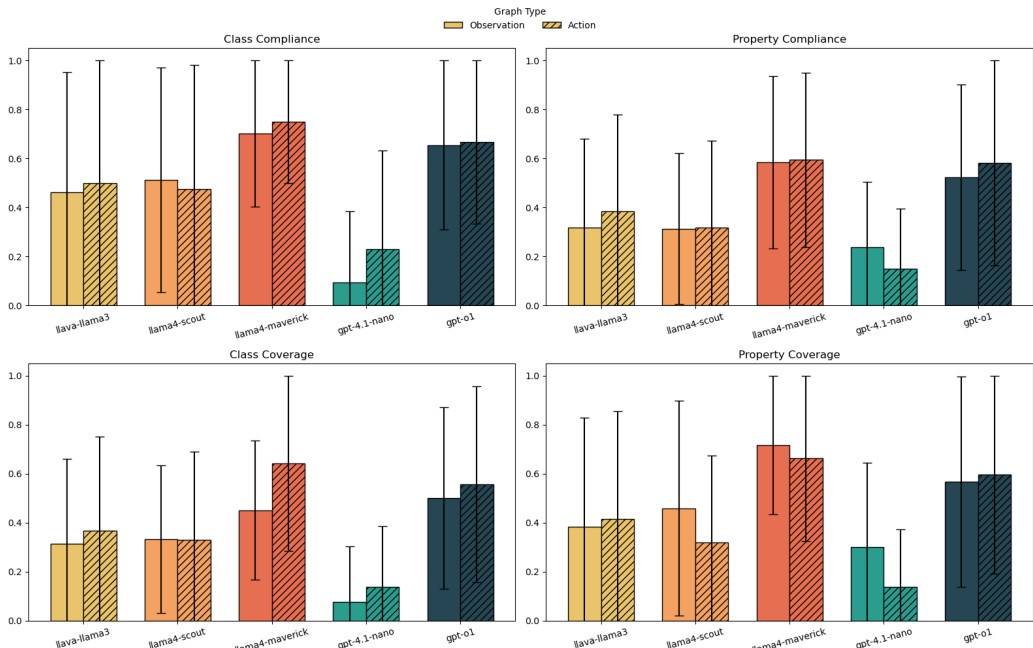

Figure 2: Compliance and coverage results across models for both observation (solid fill) and action graphs (hatched fill). Error bars represent standard deviation across 10 runs.

To further compare model performance, we generated a heatmap-like figure (Figure 3) summarizing key metrics averaged over the 10 runs. Colors range from pale yellow (low values) to blue (high values), except for the SHACL violation ratio, where lower values are better and thus mapped to blue. The shown metrics include: number of valid RDF-TTL KGs; average triple count per KG; SHACL violation per triple; compliance and coverage for classes (C), properties (P) and their averages (AVG).

These findings align with those from Figure 2, highlighting that LLaMA 4 Maverick and GPT-o1 generally outperform other models. Notably, GPT-o1 generates KGs with significantly higher triple counts (up to 332 triples), while both Maverick and GPT-o1 yield lower SHACL violation ratios—indicating stronger adherence to the ontology. Some cells in the SHACL violation column are white, signifying that the metric could not be computed because the corresponding KGs had zero compliance or coverage, i.e. they did not follow the OntoBOT ontology at all. Several additional patterns are noteworthy. All LLaMA-based models show zero compliance and coverage when using the 'dpe' (Dynamic Path Extractor) method. This suggests that LLaMA's built-in entity-relation extraction mechanism struggles with structured KG generation, given an ontology as a schema. Similarly, the 'i2kg' method (image-to-KG) using `LLaVA + LLaMa 3` fails to produce valid KGs, likely due to limitations in the vision model's capacity to handle ontology-constrained generation tasks. Further, the 'd2kg-rag' method (description-to-KG with retrieval-augmented generation) also fails to generate valid KGs for GPT-based models, across both observation and action graphs. This is unexpected, as RAG techniques are often assumed to enhance contextual grounding. However, this outcome may reflect differences in how each API handles ontology-based retrieval and response generation. In contrast, 'd2kg-rag' performs

reasonably well with other models, suggesting a possible implementation or configuration discrepancy in GPT's handling of RAG over structured knowledge, compared to the one from LLaMa-index.

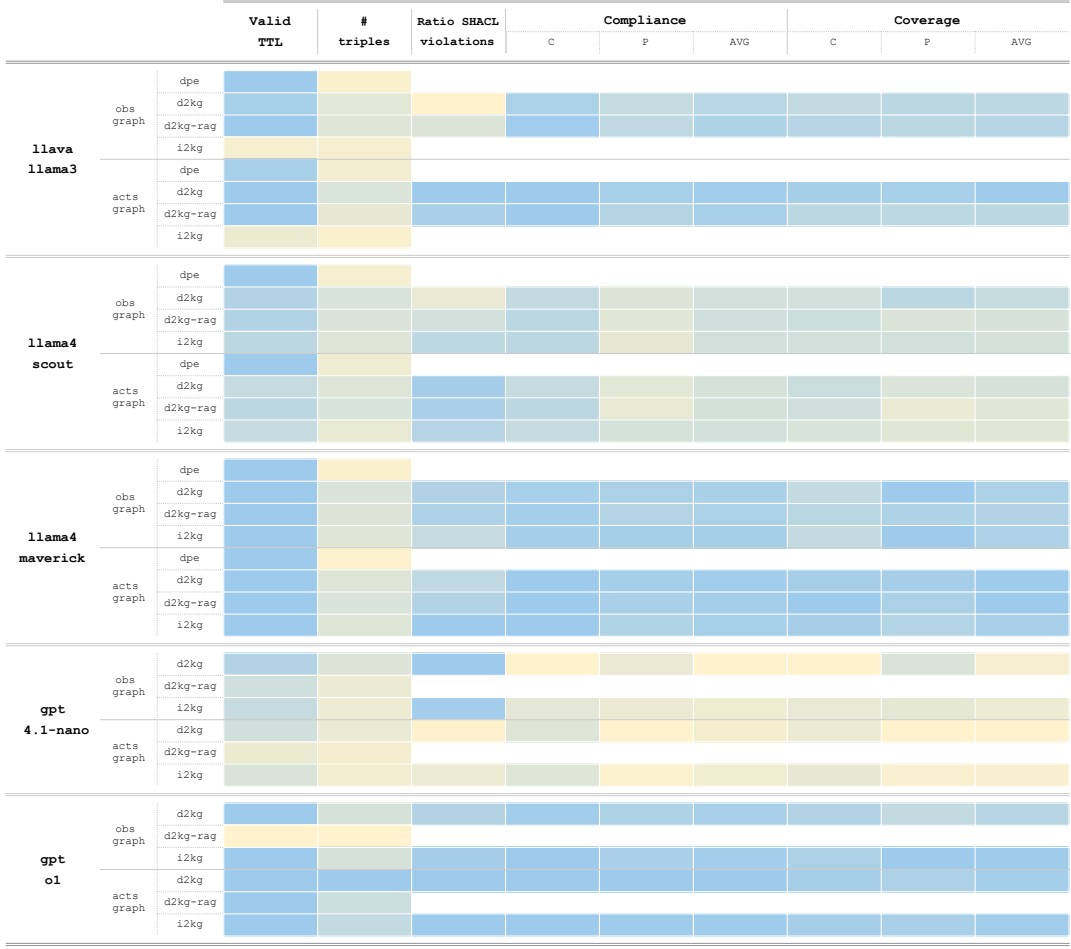

Figure 3: Heatmap comparing models across several metrics. Blue indicates better performance; yellow indicates lower performance. White cells denote metrics that could not be computed due to lack of valid RDF output.

## 4.2. Statistical Analysis

To assess whether the differences in compliance and coverage across models were statistically significant, we conducted pairwise comparisons using the Mann–Whitney U test. This non-parametric test is appropriate for comparing independent samples without assuming normality, and was applied to the median values of each metric across all runs. Here we highlight the key significant results, while a full summary of the pairwise tests is available in the Appendix (Table 1).

The test results align closely with the trends observed in Figure 2 and Figure 3. Both GPT-o1 and LLaMA 4 Maverick consistently scored higher in compliance and coverage,

with statistically significant differences found in nearly all their comparisons. However, no significant difference was detected between GPT-o1 and Maverick themselves, suggesting that both models achieve similarly performance in generating ontology-compliant KGs. Interestingly, a significant difference emerged between Maverick and Scout, despite both being LLaMA 4 models. This was somewhat unexpected, and may point to differences in fine-tuning objectives, training datasets, or system-level configurations. Another surprising finding was the lack of a statistically significant difference between LLaMA 4 Scout and the older `LLaVA + LLaMA 3` configuration. This suggests that, for KG generation tasks, newer architectures or unified pipelines do not automatically translate into better performance unless explicitly optimized for ontology-based generation.

Overall, our findings indicate that adopting a neurosymbolic approach can help generate structured, ontology-compliant, and shared symbolic representations, with the potential to facilitate more adaptable and interoperable knowledge and reasoning in robotic applications. Identifying configurations that result more consistent and structured outputs may contribute to the integration of systems across robotic platforms, addressing rigidity challenges in current architectures.

## 5. Conclusion

This study investigated how different multimodal models generate structured KGs from raw sensory inputs using a shared ontology, illustrated by a robotic use-case. We assessed outputs across compliance, coverage, and structural validity, evaluating both observation and action graphs across several neurosymbolic integration strategies. LLaMA 4 Maverick and GPT-o1 generally produced more consistent and ontology-compliant graphs compared to other models. The results also point to several nuances: for example, LLaMa 4 Scout, despite being a newer model, did not significantly outperform the older `LLaVA + LLaMA 3` setup, suggesting that architectural improvements alone may not guarantee better ontology-grounded KG generation. Moreover, strategies like 'dpe', 'i2kg', and 'd2kg-rag' were highly sensitive to model architecture, with some configurations failing entirely to produce valid outputs. These findings highlight the importance of both model design and integration strategy when aiming to generate structured, ontology-compliant outputs from raw sensory inputs. While the best-performing models show promising capabilities, variability and structural errors remain a concern, pointing to the need for further refinement, particularly in aligning generation of structured outputs through neural models. While further work is needed to improve robustness and generalization of neurosymbolic in robotic applications, this study contribute toward building more interoperable robotic architecture, where shared knowledge representation can facilitate reasoning and coordination across different environments and platforms.

### Acknowledgements

This work is funded by the Technology Exchange Programme of the Horizon Europe eu-ROBIN project (grant agreement No 101070596). Further, we acknowledge that ChatGPT was utilized to generate and debug part of the Python and LaTeXcode used in this work.

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

## Appendix A. Prompt Templates Examples

```
Ontology as context information is below.
---------------------
{ontology_ttl}
---------------------

You are an intelligent assistant tasked to generate a **Knowledge Graph of the
    environment described as follows**:
---------------------
ENVIRONMENT DESCRIPTION: {description_txt}
---------------------

Instructions:
- Analyze the description carefully to understand the complete layout of the
    environment.
- Based on the ontology, **generate a Knowledge Graph describing the environment
    **.
- All entities and relations must conform to the structure and semantics of the
    ontology.
- **Use only classes and properties from the ontology.**
- Do **NOT invent or infer any terms or actions outside of the ontology schema.**

Output format:
- Return only the generated Knowledge Graph.
- Output only text, no extra explanations.
- Use Turtle format for the output, such as <subject> <predicate> <object> .
- Include all prefixes and namespaces at the beginning.
- Use the ex: prefix with namespace <http://example.org/data/> only for newly
    instantiated entities instantiated, such as specific actions, objects, or
    locations.
- Do not use the ex: prefix for ontology classes, properties, or schema
    definitions, those must strictly come from the provided ontology with their
    original prefixes and namespaces.
```

Listing 1: Prompt template for the 'd2kg' observation graph integration method.

```
Ontology as context information is below.
---------------------
{ontology_ttl}
---------------------

You are an intelligent assistant tasked to generate a **Knowledge Graph of the
    sequence of actions a robot must perform to accomplish the following task**:
---------------------
ROBOT TASK: {robot_task}
---------------------

You are provided with text description of an environment, below:
---------------------
ENVIRONMENT DESCRIPTION: {description_txt}
```

```
--------------------

Instructions:
- Analyze the description carefully to understand the complete layout of the
    environment.
- Based on the ontology, **generate the sequence of actions required for the robot
     to complete the task**.
- Each action is a **single, atomic, clear action**.
- **All actions, entities, and relationships must strictly follow the provided
    ontology.**
- **Use only classes and properties from the ontology.**
- Do **NOT invent or infer any terms or actions outside of the ontology schema.**
- The graph should represent actions, objects involved, and their relations
    according to the ontology's structure and semantics.

Output format:
- Return only the generated Knowledge Graph of actions.
- Output only text, no extra explanations.
- Use Turtle format for the output, such as <subject> <predicate> <object> .
- Include all prefixes and namespaces at the beginning.
- Use the ex: prefix with namespace <http://example.org/data/> only for newly
    instantiated entities instantiated, such as specific actions, objects, or
    locations.
- Do not use the ex: prefix for ontology classes, properties, or schema
    definitions, those must strictly come from the provided ontology with their
    original prefixes and namespaces.
```

Listing 2: Prompt template for the 'd2kg' action graph integration method.

## Appendix B.  SHACL Shapes

The following SHACL rules were used to validate the structure and semantic correctness of the generated knowledge graphs, based on the OntoBOT ontology.

### B.1.  Shapes for observation graphs.

```
@prefix sh:     <http://www.w3.org/ns/shacl#> .
@prefix dul:    <http://www.ontologydesignpatterns.org/ont/dul/DUL.owl#> .
@prefix obot:   <https://w3id.org/onto-bot#> .

obot:EnvironmentShape a sh:NodeShape ;
    sh:targetClass obot:Environment ;
    sh:property [
        sh:path dul:hasComponent ;
        sh:or (
            [ sh:class obot:Component ]
            [ sh:class obot:Appliance ]
            [ sh:class obot:Furniture ]
            [ sh:class obot:Object ]
            [ sh:class obot:Environment]
        ) ;
        sh:minCount 1 ;
    ] .
```

Listing 3: SHACL shape for obot:Environment

```
@prefix sh:     <http://www.w3.org/ns/shacl#> .
@prefix dul:    <http://www.ontologydesignpatterns.org/ont/dul/DUL.owl#> .
@prefix soma:   <http://www.ease-crc.org/ont/SOMA.owl#> .
@prefix obot:   <https://w3id.org/onto-bot#> .

obot:ComponentShape a sh:NodeShape ;
    sh:targetClass obot:Component ;
    sh:property [
        sh:path obot:hasAffordance ;
        sh:or (
            [ sh:class obot:Affordance ]
            [ sh:class soma:Closing ]
            [ sh:class soma:Opening ]
            [ sh:class soma:Delivering ]
            [ sh:class soma:Holding ]
            [ sh:class soma:PickingUp ]
            [ sh:class soma:PuttingDown ]
            [ sh:class soma:Pulling ]
            [ sh:class soma:Pushing ]
            [ sh:class soma:Grasping ]
        ) ;
    ] ;
    sh:property [
        sh:path dul:hasLocation ;
        sh:nodeKind sh:BlankNodeOrIRI;
        sh:minCount 1 ;
    ] .
```

Listing 4: SHACL shape for obot:Component

```
@prefix sh:     <http://www.w3.org/ns/shacl#> .
@prefix geo:    <http://www.opengis.net/ont/geosparql#> .
@prefix obot:   <https://w3id.org/onto-bot#> .

obot:LocationShape a sh:NodeShape ;
    sh:targetClass obot:Location ;
    sh:property [
        sh:path obot:onTopOf ;
        sh:or (
            [ sh:class obot:Component ]
            [ sh:class obot:Appliance ]
            [ sh:class obot:Furniture ]
            [ sh:class obot:Object ]
        ) ;
    ] ;
    sh:property [
        sh:path geo:sfContains ;
        sh:or (
            [ sh:class obot:Component ]
            [ sh:class obot:Appliance ]
            [ sh:class obot:Furniture ]
            [ sh:class obot:Object ]
        ) ;
    ] ;
    sh:property [
        sh:path geo:sfWithin ;
        sh:or (
            [ sh:class obot:Component ]
            [ sh:class obot:Appliance ]
```

```
            [ sh:class obot:Furniture ]
            [ sh:class obot:Object ]
        ) ;
    ] ;
    sh:property [
        sh:path geo:sfOverlaps ;
        sh:or (
            [ sh:class obot:Component ]
            [ sh:class obot:Appliance ]
            [ sh:class obot:Furniture ]
            [ sh:class obot:Object ]
        ) ;
    ] .
```

Listing 5: SHACL shape for obot:Location

## B.2. Shapes for action graphs.

```
@prefix sh:      <http://www.w3.org/ns/shacl#> .
@prefix obot:    <https://w3id.org/onto-bot#> .

obot:InstructionShape a sh:NodeShape ;
    sh:targetClass obot:Instruction ;
    sh:property [
        sh:path obot:hasWorkflow ;
        sh:class obot:Workflow ;
    ] ;
    sh:property [
        sh:path obot:hasNaturalLanguage ;
    ] .
```

Listing 6: SHACL shape for obot:Instruction

```
@prefix sh:      <http://www.w3.org/ns/shacl#> .
@prefix dul:     <http://www.ontologydesignpatterns.org/ont/dul/DUL.owl#> .
@prefix obot:    <https://w3id.org/onto-bot#> .

obot:WorkflowShape a sh:NodeShape ;
    sh:targetClass obot:Workflow ;
    sh:property [
        sh:path obot:hasAction ;
        sh:qualifiedValueShape [
            sh:class dul:Action ;
        ] ;
        sh:qualifiedMinCount 1 ;
    ] .
```

Listing 7: SHACL shape for obot:Workflow

```
@prefix sh:      <http://www.w3.org/ns/shacl#> .
@prefix dul:     <http://www.ontologydesignpatterns.org/ont/dul/DUL.owl#> .
@prefix soma:    <http://www.ease-crc.org/ont/SOMA.owl#> .
@prefix obot:    <https://w3id.org/onto-bot#> .

obot:ActionShape a sh:NodeShape ;
    sh:targetClass dul:Action ;
    sh:or (
        [ sh:property [
            sh:path dul:precedes ;
        ] ]
        [ sh:targetSubjectsOf dul:precedes ]
```

```
        [ sh:property [
            sh:path dul:follows ;
        ] ]
        [ sh:targetSubjectsOf dul:follows ]
    ) ;
    sh:property [
        sh:path soma:isPerformedBy ;
    ] ;
    sh:property [
        sh:path obot:isAffordedBy ;
        sh:minCount 1 ;
    ] ;
    sh:property [
        sh:path obot:actsOn ;
        sh:minCount 1 ;
    ] .
```

Listing 8: SHACL shape for dul:Action

## Appendix C. Generated KGs

```
@prefix owl: <http://www.w3.org/2002/07/owl#> .
@prefix geo: <http://www.opengis.net/ont/geosparql#> .
@prefix soma: <http://www.ease-crc.org/ont/SOMA.owl#> .
@prefix dul: <http://www.ontologydesignpatterns.org/ont/dul/DUL.owl#> .
@prefix rdfs: <http://www.w3.org/2000/01/rdf-schema#> .
@prefix obot: <https://w3id.org/onto-bot#> .
@prefix ex: <http://example.org/data/> .

ex:Kitchen a obot:Environment ;
    dul:hasComponent ex:Countertop1, ex:Fridge, ex:DiningTable, ex:Cabinet1, ex:
    Cabinet2, ex:ExhaustHood, ex:Sink .
ex:Countertop1 a obot:Component ;
    obot:hasAffordance soma:Holding, soma:PickingUp, soma:PuttingDown ;
    dul:hasLocation ex:Kitchen ;
    geo:sfContains ex:Jar1, ex:Jar2, ex:Box1, ex:Box2, ex:Plates .
ex:Jar1 a obot:Object ;
    dul:hasLocation ex:Countertop1 .
ex:Jar2 a obot:Object ;
    dul:hasLocation ex:Countertop1 .
ex:Box1 a obot:Object ;
    dul:hasLocation ex:Countertop1 .
ex:Box2 a obot:Object ;
    dul:hasLocation ex:Countertop1 .
ex:Plates a obot:Object ;
    dul:hasLocation ex:Countertop1 .
ex:Fridge a obot:Appliance ;
    obot:hasAffordance soma:Opening, soma:Closing ;
    dul:hasLocation ex:Kitchen .
ex:DiningTable a obot:Furniture ;
    obot:hasAffordance soma:Holding, soma:PickingUp, soma:PuttingDown ;
    dul:hasLocation ex:Kitchen ;
    geo:sfContains ex:WaterBottle, ex:Book1 .
```

```
ex:WaterBottle a obot:Object ;
    dul:hasLocation ex:DiningTable .
ex:Book1 a obot:Object ;
    dul:hasLocation ex:DiningTable .
ex:Cabinet1 a obot:Furniture ;
    obot:hasAffordance soma:Opening, soma:Closing ;
    dul:hasLocation ex:Kitchen .
ex:Cabinet2 a obot:Furniture ;
    obot:hasAffordance soma:Opening, soma:Closing ;
    dul:hasLocation ex:Kitchen .
ex:ExhaustHood a obot:Appliance ;
    dul:hasLocation ex:Kitchen .
ex:Sink a obot:Component ;
    obot:hasAffordance soma:Holding ;
    dul:hasLocation ex:Countertop1 .
ex:Countertop1 dul:hasLocation ex:Kitchen ;
    obot:onTopOf ex:Cabinet1 .
ex:DiningTable dul:hasLocation ex:Kitchen .
ex:Fridge dul:hasLocation ex:Kitchen .
ex:Cabinet1 dul:hasLocation ex:Kitchen .
ex:Cabinet2 dul:hasLocation ex:Kitchen .
ex:ExhaustHood dul:hasLocation ex:Kitchen .
ex:Sink dul:hasLocation ex:Countertop1 .
```

Listing 9: Observation graph generated by LLaMa 4 Maverick through the 'i2kg' method.

```
@prefix obot:    <https://w3id.org/onto-bot#> .
@prefix dul:     <http://www.ontologydesignpatterns.org/ont/dul/DUL.owl#> .
@prefix soma:    <http://www.ease-crc.org/ont/SOMA.owl#> .
@prefix xsd:     <http://www.w3.org/2001/XMLSchema#> .
@prefix :        <http://example.org/kb#> .

:Instruction_1 a obot:Instruction ;
    obot:hasNaturalLanguage "Restore the kitchen to an organized state by
    identifying all misplaced items and returning them to their standard storage
    locations based on their type and function. Prioritize actions according to
    logical task order, and perform each step atomically." ;
    obot:hasWorkflow :Workflow_1 .
:Workflow_1 a obot:Workflow ;
    obot:hasAction :Action_1, :Action_2, :Action_3, :Action_4, :Action_5, :
    Action_6,
                  :Action_7, :Action_8, :Action_9, :Action_10, :Action_11, :
    Action_12,
                  :Action_13, :Action_14 .
:Robot a dul:Agent .
:Fridge a obot:Appliance .
:Cabinet a obot:Furniture .
:Jam a obot:Object .
:WaterBottle a obot:Object .
:Cereal a obot:Object .
:Plates a obot:Object .
```

```
:Pot a obot:Object .
:Action_1 a dul:Action ;
    soma:isPerformedBy :Robot ;
    obot:actsOn :Jam ;
    obot:isAffordedBy soma:PickingUp ;
    obot:precedes :Action_2 .
:Action_2 a dul:Action ;
    soma:isPerformedBy :Robot ;
    obot:actsOn :Fridge ;
    obot:isAffordedBy soma:Opening ;
    obot:follows :Action_1 ;
    obot:precedes :Action_3 .
:Action_3 a dul:Action ;
    soma:isPerformedBy :Robot ;
    obot:actsOn :Jam ;
    obot:isAffordedBy soma:PuttingDown ;
    obot:follows :Action_2 ;
    obot:precedes :Action_4 .
:Action_4 a dul:Action ;
    soma:isPerformedBy :Robot ;
    obot:actsOn :WaterBottle ;
    obot:isAffordedBy soma:PickingUp ;
    obot:follows :Action_3 ;
    obot:precedes :Action_5 .
:Action_5 a dul:Action ;
    soma:isPerformedBy :Robot ;
    obot:actsOn :WaterBottle ;
    obot:isAffordedBy soma:PuttingDown ;
    obot:follows :Action_4 ;
    obot:precedes :Action_6 .
:Action_6 a dul:Action ;
    soma:isPerformedBy :Robot ;
    obot:actsOn :Fridge ;
    obot:isAffordedBy soma:Closing ;
    obot:follows :Action_5 ;
    obot:precedes :Action_7 .
:Action_7 a dul:Action ;
    soma:isPerformedBy :Robot ;
    obot:actsOn :Cereal ;
    obot:isAffordedBy soma:PickingUp ;
    obot:follows :Action_6 ;
    obot:precedes :Action_8 .
:Action_8 a dul:Action ;
    soma:isPerformedBy :Robot ;
    obot:actsOn :Cabinet ;
    obot:isAffordedBy soma:Opening ;
    obot:follows :Action_7 ;
    obot:precedes :Action_9 .
:Action_9 a dul:Action ;
    soma:isPerformedBy :Robot ;
    obot:actsOn :Cereal ;
```

```
      obot:isAffordedBy soma:PuttingDown ;
      obot:follows :Action_8 ;
      obot:precedes :Action_10 .
:Action_10 a dul:Action ;
      soma:isPerformedBy :Robot ;
      obot:actsOn :Plates ;
      obot:isAffordedBy soma:PickingUp ;
      obot:follows :Action_9 ;
      obot:precedes :Action_11 .
:Action_11 a dul:Action ;
      soma:isPerformedBy :Robot ;
      obot:actsOn :Plates ;
      obot:isAffordedBy soma:PuttingDown ;
      obot:follows :Action_10 ;
      obot:precedes :Action_12 .
:Action_12 a dul:Action ;
      soma:isPerformedBy :Robot ;
      obot:actsOn :Pot ;
      obot:isAffordedBy soma:PickingUp ;
      obot:follows :Action_11 ;
      obot:precedes :Action_13 .
:Action_13 a dul:Action ;
      soma:isPerformedBy :Robot ;
      obot:actsOn :Pot ;
      obot:isAffordedBy soma:PuttingDown ;
      obot:follows :Action_12 ;
      obot:precedes :Action_14 .
:Action_14 a dul:Action ;
      soma:isPerformedBy :Robot ;
      obot:actsOn :Cabinet ;
      obot:isAffordedBy soma:Closing ;
      obot:follows :Action_13 .
```

Listing 10: Action graph generated by GPT-o1 through the 'i2kg' method.

## Appendix D. Statistics

Table 1: Pairwise model comparisons for Compliance and Coverage. Asterisks indicate statistically significant differences ($p < 0.05$).

| Metric | Model 1 | Model 2 | Median 1 / 2 | $p$-value | Sig. |
|---|---|---|---|---|---|
| Compliance | gpt-4.1-nano | llava-llama3 | 0.000 / 0.000 | 0.003 | * |
| | | llama4-scout | 0.000 / 0.514 | 0.004 | * |
| | | llama4-maverick | 0.000 / 0.857 | <001 | * |
| | | gpt-o1 | 0.000 / 0.900 | <001 | * |
| | gpt-o1 | llava-llama3 | 0.900 / 0.000 | <001 | * |
| | | llama4-scout | 0.900 / 0.514 | <001 | * |
| | | llama4-maverick | 0.900 / 0.857 | 0.271 | |
| | llama4-maverick | llava-llama3 | 0.857 / 0.000 | <001 | * |
| | | llama4-scout | 0.857 / 0.514 | <001 | * |
| | llama4-scout | llava-llama3 | 0.514 / 0.000 | 0.302 | |
| Coverage | gpt-4.1-nano | llava-llama3 | 0.000 / 0.000 | 0.005 | * |
| | | llama4-scout | 0.000 / 0.417 | 0.002 | * |
| | | llama4-maverick | 0.000 / 0.750 | <001 | * |
| | | gpt-o1 | 0.000 / 0.854 | <001 | * |
| | gpt-o1 | llava-llama3 | 0.854 / 0.000 | 0.007 | * |
| | | llama4-scout | 0.854 / 0.417 | <001 | * |
| | | llama4-maverick | 0.854 / 0.750 | 0.344 | |
| | llama4-maverick | llava-llama3 | 0.750 / 0.000 | <001 | * |
| | | llama4-scout | 0.750 / 0.417 | <001 | * |
| | llama4-scout | llava-llama3 | 0.417 / 0.000 | 0.478 | |

