# OpenReview forum: "Bridging Bots: from Perception to Action via Multimodal-LMs and Knowledge Graphs"
_nesyconf.org/NeSy/2025/Conference_Phase_2 — NeSy 2025 - Phase 2 Poster_

### Official Review · Reviewer_eFhn · 2025-07-01
**Not a Technical Paper, not a Strong Application Paper**

**Rating:** 4
**Confidence:** 5

**Review:**

The paper proposes the use of multimodal language models to generate the parsing of a scene together with an action graph describing the actions to achieve a goal in the scene. The goal is provided as natural language text in the input. The input also includes an ontology. Both the scene and the action graph have to be compliant with the ontology. Several multimodal LLMs have been tested.

This is not a technical paper but an application paper where the NeSy integration relies in the mixed types of input: pictures, task description and an ontology (symbolic part). All the computations are performed by multimodal LLMs with accurate prompts. The evaluation assesses the quality of the generated graphs. This makes the work an application paper to a synthetic scenario as the images are generated by a simulation platform. In addition, there is no description of the dataset. The paper could be worth to be published if applied to a real scenario. This synthetic setting gives just an initial evaluation of the approach and is more suitable for a workshop.

Other concerns
- Section 2.1, please cite early works for scene parsing with ontology alignment, such as, this https://journals.sagepub.com/doi/abs/10.3233/IA-160093
- Section 2, the state-of-the-art should include a section with works performing planning with multimodal LLMs as the action graph represents a plan.
- Page 4, on out framework -> on our framework
- How is the ontology encoded in the prompt? As triples or is there a more sophisticated serialization in natural language? Please specify.
- Page 5, and assess -> and assesses
- The experiments should include ablation studies where the ontology is not used. Otherwise it is not possible to understand whether the compliance is due to the ontology or to the information inside the LLM,

**Anonymity:**

Disclose identity

---

### Official Review · Reviewer_mu3E · 2025-07-08
**An LM-based Neurosymbolic Pipeline for Ontology-compliant Knowledge Graph Generation in Service Robots**

**Rating:** 6
**Confidence:** 3

**Review:**

The paper investigates the application of a neurosymbolic pipeline based on Language Models to generate Ontology-compliant knowledge graphs for service robots. This pipeline integrates raw sensory data, natural language task descriptions, ontologies and multi-modal language models to generate both observation and action graphs. Experiments are conducted with 4 different neurosymbolic methods and 5 different LMs, showing how the choice of LM can greatly affect the quality of the generated graphs, and how some LMs show promising results, but still present limitations.

Quality: good: the paper presents a technically sound method, further supported by the fact that it mostly reuses existing work.

Clarity: very good: the presentation of the paper content is very clear. The presented ideas are laid out in a sensible and predictable structure. Language is clear, concise, and easily readable. Figures and tables are clear and easily interpretable.

Originality: low: the proposed pipeline combines existing neurosymbolic methodologies applying them to service robots, meaning originality only lies in the application and combination of the used method(s).

Significance: good: the method is studied empirically on a relevant service robot domain. Experiments are comprehensive as both the LM and neurosymbolic methods are varied. Results are significant and appropriately reflect their presented discussion. Reproducibility is supported by the full code provided in an online repository.

Strengths:
- the paper is technically sound
- the presentation is clear, concise, and comprehensive
- experiments are comprehensive and results are significant

Weaknesses:
- originality: the proposed pipeline only combines existing methods/models, applying them to service robots
- comparison between different neurosymbolic methods is not explicit and hard to visualize, although this would require no additional experiments (see below)

Detailed comments:
- While figure 3 is well suited to present a comparison between different LMs, it is hard to visually compare different neurosymbolic methods. Adding a similar figure with neurosymbolic methods as the outer (leftmost) column would allow a more immediate comparison, possibly warranting a more detailed discussion. This could also be included in the appendix, with a direct reference to it whenever figure 3 is mentioned and in its caption.
- The specific domain used for the experiments (Webots) is only mentioned once in section 3.1. Mentioning it again at the start of the experimental sections, along with a brief description, would allow readers to more easily find this piece of information.
- Directly adding citations to the bullet lists of neurosymbolic methods in 3.2 would better direct readers to the respective works

**Anonymity:**

Remain anonymous

---

### Official Review · Reviewer_2a3J · 2025-07-08

**Rating:** 6
**Confidence:** 2

**Review:**

This research proposes a neurosymbolic framework that combines VLMs with knowledge graphs for robotic applications.
The framework takes as input multiple images of a scene and a task description, and outputs two knowledge graphs that includes objects, actions.
The authors evaluate multiple VLMs, and found that GPT-o1 and LLaMA 4 Maverick produced more consistent and accurate knowledge graphs.
However, the results also demonstrated that newer models such as LLaMA 4 Scout did not performed better than older models such as LLaVA + LLaMA 3.

The research is well-motivated, and the proposed framework is interesting.
Studying the performance of VLMs in generating knowledge graphs for robotic applications is a relevant open problem.
The paper validates multiple VLMs, and knowledge graph generation strategies.

Some elements that could be improved are:
* Further explanations in section 3.2 of how the KG generation is performed would substantially increase the clarity of the paper.
* It's not clear how many scenes or images per scene the authors used to prompt the VLMs.
* Moreover, if one single simulated scene is used, it might not provide a good estimate of the performance of the system in real world settings.
* For the evaluation, 10 runs would often not be enough to get a good estimate of the performance.
* Running each experiment at least 20 times and report the mean and standard deviation would be a standard practice.
* More in-depth discussion of why newer models might not outperform older ones.

**Anonymity:**

Remain anonymous